# Discriminating between negative cooperativity and ligand binding to independent sites using pre-equilibrium properties of binding curves

**Federico Sevlever**[1,2], **Juan Pablo Di Bella**[1,2], **Alejandra C. Ventura**[1,2]*

**1** Department of Physiology, Molecular and Cellular Biology, University of Buenos Aires, Buenos Aires, Argentina, **2** Institute of Physiology, Molecular Biology and Neurosciences, National Research Council (CONICET), Buenos Aires, Argentina

* alejvent@fbmc.fcen.uba.ar

**Data Availability Statement:** All relevant data are within the manuscript and its Supporting Information files.

## Abstract

Negative cooperativity is a phenomenon in which the binding of a first ligand or substrate molecule decreases the rate of subsequent binding. This definition is not exclusive to ligand-receptor binding, it holds whenever two or more molecules undergo two successive binding events. Negative cooperativity turns the binding curve more graded and cannot be distinguished from two independent and different binding events based on equilibrium measurements only. The need of kinetic data for this purpose was already reported. Here, we study the binding response as a function of the amount of ligand, at different times, from very early times since ligand is added and until equilibrium is reached. Over those binding curves measured at different times, we compute the dynamic range: the fold change required in input to elicit a change from 10 to 90% of maximum output, finding that it evolves in time differently and controlled by different parameters in the two situations that are identical in equilibrium. Deciphering which is the microscopic model that leads to a given binding curve adds understanding on the molecular mechanisms at play, and thus, is a valuable tool. The methods developed in this article were tested both with simulated and experimental data, showing to be robust to noise and experimental constraints.

## Author summary

When two successive events occur, it may make sense to know if they affect somehow each other, particularly if the properties of the second event are modified by the occurrence of the first one. Two scenarios lead to the same overall outcome: first, the two events are identical but they interfere with each other, and second, the two events are independent but non identical. The interference caused in the first scenario produces the same result as having a second event with different properties. Now, let's name these events as bindings, the interference as negative cooperativity, and the non-identical events as independent binding. In this work we focus on the dynamic process by which the two

**Funding:** This work was supported by a grant from the from the Argentine Agency of Research and Technology (PICT2016-0130) to ACV (http://www.agencia.mincyt.gob.ar/frontend/agencia/instrumento/24). The funders had no role in study design, data collection and analysis, decision to publish, or preparation of the manuscript.

**Competing interests:** The authors have declared that no competing interests exist.

scenarios produce the same result. We selected a relevant but not characterized before property of the binding process, called its dynamic range, and found it behaves differently in these two scenarios and controlled by different parameters of interest. Based on this feature, we developed and algorithm to distinguish between negative cooperativity and independent binding based on the time evolution of the dynamic range. This tool allows to discover the microscopic model behind the data and may be useful in other similar problems in cell signaling.

## Introduction

Cells detect input signaling molecules using receptors, proteins usually located on the cell surface embedded in the plasma membrane. Activated receptors then transmit the signal to the interior of the cell through a series of downstream processes that typically lead to changes in gene expression, resulting in an appropriate output response to the input. In a way, the system's overall input-output curve summarizes its biological characteristics and function [1].

Receptors have usually more than one binding site. Cooperativity in binding is defined as a change in the properties of a given site depending on the state (occupied or not) of the other. For two identical sites, if the second binding is weaker once the first site is occupied, this is called negative cooperativity. The opposite corresponds to positive cooperativity.

Cooperativity is widely spread in biological systems and has an important role in regulating signaling responses [2,3], particularly, cooperative interactions are used to accelerate or otherwise enhance specific processes [3]. It usually arises from allosteric communication between the binding sites, but not exclusively. Enforced proximity or the avidity effect was described in protein-protein interactions and could lead to cooperativity without allosteric communication [2]. Cooperativity not only occurs during ligand binding; it can also happen in other processes involving multiple units, such as the folding and unfolding of proteins, as well as the melting of phospholipid chains that comprise the cell membrane. In these cases cooperativity can be macroscopically understood by analogy with a first-order phase transition [4]. Positive cooperativity in folding of proteins means that a second folding is more likely to occur than a first one, a third folding more likely to occur than a second, and so on. This leads to a critical point in the variation of some control parameter (usually temperature) where the folding changes abruptly, giving an all-or-none response. The relation between cooperativity and first-order phase transitions is also found in different systems, from interactive atoms [5] to epidemics modeling [6]. Another example where cooperativity arises is the unwinding of DNA. Sections of DNA must first unwind in order for the DNA to carry out its other functions, such as replication, transcription, and recombination. Positive cooperativity among adjacent DNA nucleotides makes it easier to unwind the whole group than it is to unwind the same number of nucleotides spread out along the DNA chain. However, enzymes are also involved in this example, so binding is not completely excluded. There are also examples of membrane-derived cooperativity and cooperative interactions between cells. Cooperativity has also shown to have an important role in the area of drug discovery [3]. Cooperativity, even when it was recognized as a critical enabling mechanism for life, remains poorly understood.

Positive cooperativity produces an input-output response that looks switch-like, in the sense that low input produces no significant output until reaching a threshold, while inputs greater than the threshold produce almost maximal output. A typical example is the binding of oxygen to hemoglobin [7]. Negative cooperativity, on the other hand, gives an input-output curve that is more graded than the curve with no cooperativity. However, it has recently been

shown that if ligand depletion is considered, negative cooperativity can also produce a marked threshold in the input-output curve [8]. Despite the fact that negative cooperativity is almost as common as positive, it has received less attention. More than 45 years ago, A. Cornish-Bowden wrote an article highlighting the physiological significance of negative cooperativity [9]. Years have passed and the physiological role of negative cooperativity has remained unclear [10]. Al least, what became clear is that negative cooperativity is not a rare event that occurs in a few enzymes with unusual structural features, on the contrary, it is an ubiquitous feature of enzymes [11] and it is relevant in a variety of processes. To mention some of them, the advantages of negative cooperativity were characterized in metabolic systems [12] using computational models in the context of an inhibitor binding to an enzyme. On a different context, negative cooperativity has shown to play an important role in tuning transcriptional regulation. The prevalence of intrinsic disorder and multivalency among transcription factors suggests that formation of heterogeneous, dynamic complexes is a widespread mechanism, and negative cooperativity is an important feature in this scenario [13]. Finally, a direct correlation between negative cooperativity and receptor oligomerization was reported for a particular subfamily of GPCR [14]. These studies together with the accumulating evidence that most, if not all, GPCRs may oligomerize lead to the speculation that negative cooperativity is a general phenomenon in, at least, some subfamilies of GPCRs.

Negative cooperativity is not readily distinguishable from ligand binding to multiple independent sites present on a given (macro)molecule, each with different affinity. However, this indistinguishability only happens at the equilibrium states, the systems are not identical over the complete time courses of the binding reaction. As far as we know, the oldest references in the literature related to the kinetic differentiation between negative cooperativity and independent binding are due to Malatesta and Ascenzi [15] and to Wang and Pan [16]. In the first reference [15], a very short article from 1994, the indistinguishability is stated together with an example, and it is mentioned with no proof that it may be possible to discriminate between the two ligand binding mechanisms only from the kinetic viewpoint. The second reference [16], proposed a kinetic method to distinguish between these two possible binding mechanisms from the closed-form solution of the differential equations obtained under two restrictive approximations: irreversible binding and pseudo-first order for the ligand association step. In more recent years, two groups have added valuable works into the kinetic differentiation approach. One of them showed that kinetics can be used to distinguish between different molecules with the same equilibrium distribution, based on a stochastic model as well as on a deterministic one [17]. The other one used deterministic simulations and added an equation for the concentration of ligand [18]. In fact, they have previously reported an experimental study showing the efficacy of the kinetic approach in identifying two classes of binding sites for that system [19].

The focus of the current work is on studying how the dose-response curve of systems consisting of two binding sites evolves over time and how this evolution confers a tool to add on the kinetic differentiations approaches previously developed [20]. We have shown before that when a ligand-receptor system is exposed to a step-like temporal profile of ligand, the occupied receptor dose-response curve changes over time in such a way that the $EC_{50}$ (concentration of ligand that occupies 50% of the receptors) becomes progressively smaller with a minimum when the binding reaction reaches steady-state (equilibrium binding) [21]. We have called the property of systems that change their $EC_{50}$ over time, *shift*, and the systems that exhibit this property, *shifters*. We have later shown that covalent modification cycles and gene expression systems work as shifters as well [22]. In the present article we focus on independent binding and negative cooperativity, as they are both shifters but with different properties. Based on these properties, we designed an algorithm to distinguish those two situations.

What our work adds on the kinetic differentiation approaches developed before is three-fold. First, it is based on an observable that is both global and time-dependent: the dynamic range of the dose-response curve as a function of time. It is global in the sense that is extracted from the input-output curve, so it contains somehow information of the output at different inputs. These features of the selected observable make it useful for the goal of this paper and robust against noise and experimental constraints. Second, the ideas are organized as a method or algorithm, with several checkpoints, facilitating the connection between theory and experiment. Third, the proposed method is tested overall the parameter space, evaluating its performance both with simulated and experimental data.

The organization of the paper is as follows. We first study binding models with two binding sites, focusing on two microscopic schemes: independent binding and negative cooperativity with identical sites, showing that there is a manifold in the parameter space where both schemes lead to the same equilibrium dose-response curve. We then apply our *shifting* formalism focusing on the dynamic range (the fold change required in input to elicit a change from 10 to 90% of maximum output) of the dose-response curve during shifting. Based on the temporal evolution of the dynamic range we then design an algorithm able to differentiate between independent binding and negative cooperativity. We evaluate the performance of the algorithm with numerical simulations with different levels of stochasticity. Finally, we test the algorithm with experimental data. We conclude by discussing the potential implications of the results in the article.

## Results

### Binding models with two binding sites: A single equilibrium dose-response curve, two possible scenarios behind

We consider a receptor R with two binding sites for the same ligand L. From there, we focus on two scenarios, an independent binding model (**IB**) and a negative cooperativity model (**NC**). We consider different binding ($k_{ij}$) and unbinding ($l_{ij}$) rates for the two sites (01 and 10) in the **IB** model, while for the **NC** model the sites are identical, thus, characterized by a set of binding ($k$) and unbinding ($l$) rates and a cooperativity factor $\omega$, as indicated in Fig 1A. If $\omega > 1$, the second binding rate is larger than the first one, modeling positive cooperativity. If $\omega < 1$, it is smaller, resulting in negative cooperativity.

The deterministic kinetic description of the two binding models (see Methods) leads to the dose-response curves in equilibrium ($\theta$ vs. L) expressed by Eqs (1) and (2) for the **IB** and **NC** model respectively.

$$\theta_{\mathbf{IB}} = \frac{1}{2}\frac{L(K_{10} + K_{01} + 2L)}{K_{10}K_{01} + (K_{10} + K_{01})\,L\,+\,L^2} = \frac{1}{2}\left(\frac{L}{K_{10}\,+\,L} + \frac{L}{K_{01}\,+\,L}\right) \tag{1}$$

$$\theta_{\mathbf{NC}} = \frac{LK + \omega L^2}{K^2 + 2KL + \omega L^2}, \tag{2}$$

$K_{10}$ and $K_{10}$ are the dissociation constants for each binding ($K_{ij} = l_{ij}/k_{ij}$) in the **IB** model, and K and K/$\omega$ are the dissociation constants in the **NC** model (K = $l/k$). Two expressions are included for the **IB** model because in the right one it is easier to see the independency between sites, as the total expression is the average of two single binding models. The left expression results from taking a common denominator and makes it easier to compare **IB** and **NC** dose-response curves. By equaling both expressions, we find the following conditions leading to the

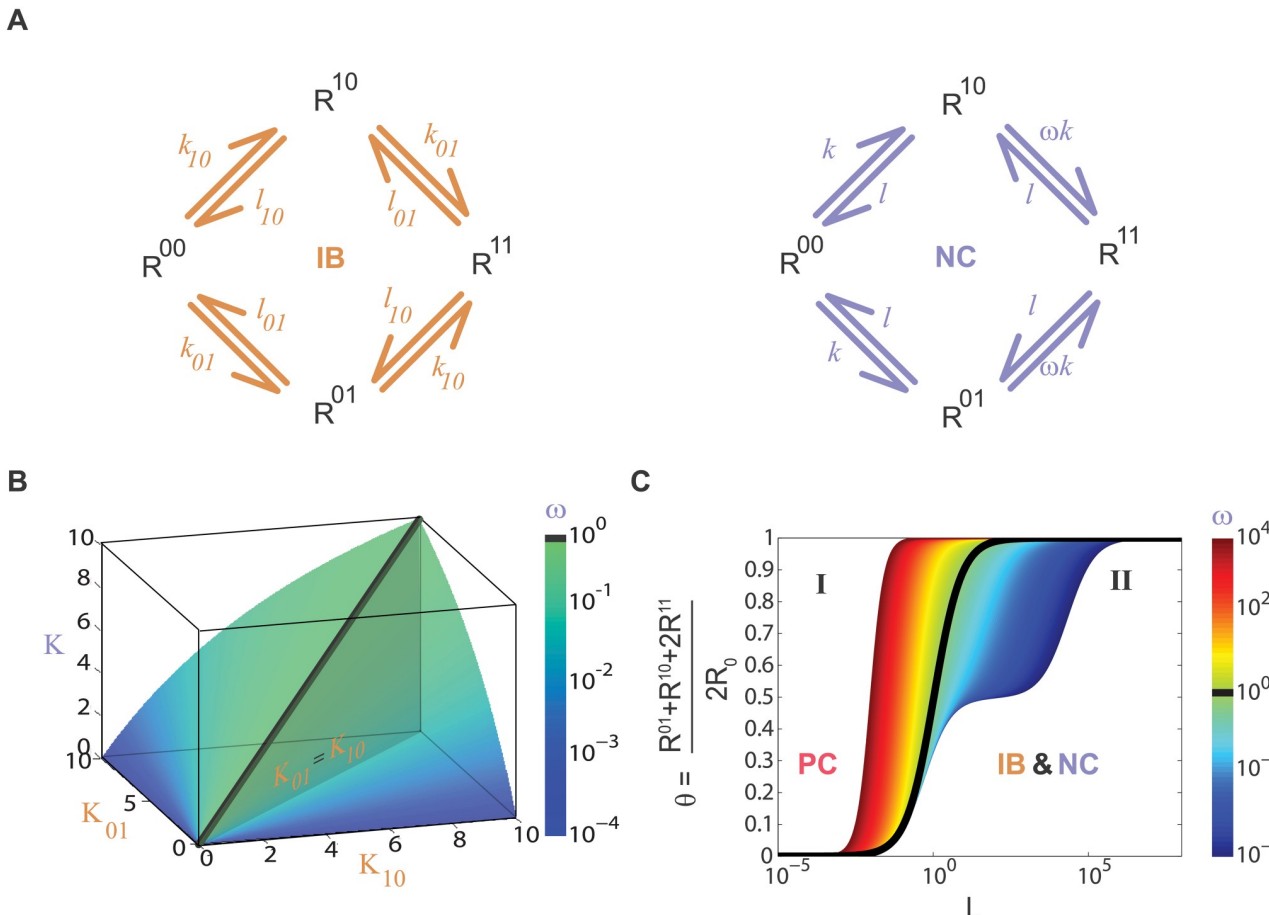

**Fig 1. Binding models with two binding sites: Independent binding and negative cooperativity leads to identical dose-response curves in equilibrium.** (A). Independent binding (**IB**, left) and negative cooperativity (**NC**, right) between a receptor R with two binding sites and a ligand L. $R^{ij}$, with i, j = 0,1, explicitly indicates if the site is empty (0) or occupied (1). **IB** has four parameters, representing binding ($k_{ij}$) and unbinding ($l_{ij}$) rates for the two different sites (01 and 10), while **NC** has three parameters, binding ($k$) and unbinding ($l$) rates of the two identical sites and ω for cooperativity. (B). Manifold where **IB** and **NC** binding models lead to identical dose-response curves in equilibrium, it is obtained from Eqs (3) and (4). One point in this plot represents a set of ($K_{10}$, $K_{01}$) from **IB** and (K, ω) from **NC**. For this panel ω < 1, ω indicated in color scale and ω = 1 is the black line. (C). **NC** and **IB** equilibrium dose-response curves for the proportion of occupied sites (**θ**), L is in log scale and K = 1 (Eqs (1) and (2)). For **NC**, there is one curve for each value of ω in color scale, the black line represents ω = 1 and separates positive cooperativity curves (region **I**) from negative cooperativity ones (region **II**). The curves in region **II** can also be obtained with the **IB** model because of the non-identifiability problem.

same equilibrium dose-response curves for **IB** and **NC** scenarios:

$$\frac{K^2}{\omega} = K_{10}K_{01} \tag{3}$$

$$\frac{K}{\omega} = \frac{K_{10} + K_{01}}{2}. \tag{4}$$

This system can be solved only if ω < 1, exhibiting an identifiability problem between **IB** and **NC** scenarios. The solutions are:

$$K_{10} = \frac{K + \sqrt{1 - \omega}}{\omega} \tag{5}$$

$$K_{01} = \frac{K - \sqrt{1 - \omega}}{\omega} \tag{6}$$

When the cooperativity factor is equal to 1 ($\omega = 1$), both roots are null and correspond to the case of identical and independent sites ($K_{10} = K_{01} = K = K/\omega$). When it is greater than 1 ($\omega > 1$), both roots are complex numbers having no physical meaning. The solution of system of Eqs (3) and (4) leads to the manifold plotted in Fig 1B. One point in this plot represents one set of ($K$, $\omega$) from **NC** and ($K_{10}$, $K_{01}$) from **IB**, which have the same equilibrium dose-response curve. Each point represents infinite different sets of **NC** or **IB** parameters, as only the dissociation constants are specified. Also, for any set of ($K$, $\omega$), there are two sets of ($K_{10}$, $K_{01}$), as a result of a reflection symmetry along the vertical plane defined by $K_{10} = K_{01}$ (vertical gray plane), which represents both model's symmetry in swapping binding sites (01↔10).

Dose-response curves are plotted in Fig 1C, with $\omega$ coded in a color scale. The curve with $\omega = 1$ (in black) divides the plot in two regions, **I** and **II**. Region **I** corresponds to positive cooperativity curves only (Eq (2), $\omega > 1$) while region **II** represents both negative cooperativity (Eq (2), $\omega < 1$) and independent binding curves (Eq (1)). These two sets of curves in region **II** are identical, meaning that they are obtained by Eqs (1) or (2), the parameters in both expressions are related by the conditions in Eqs (3) and (4). Moreover, we can use these conditions to find which set of **NC** parameters corresponds to an indistinguishable set of **IB** parameters and vice versa, defining an effective cooperativity factor for an **IB** set.

## Focusing on pre-equilibrium conditions

A dose-response curve is typically thought of in equilibrium, i.e. several doses are applied to the system and the equilibrium response is measured. The different pairs dose-response are then used to build the equilibrium dose-response curve. This curve is usually, but not always, an increasing monotonic function of the dose, until reaching saturation, as is the case for a simple ligand-binding reaction (Fig 2A). Some global quantities characterize the curve, like the $EC_{50}$ (the concentration of the dose that produces 50% of the maximal response) and the dynamic range **DynR** (the range of doses that can be distinguished according to their responses, usually calculated as indicated in Methods). This last concept is closely related to the span in free ligand concentration introduced by G. Weber in 1965 [23]. If the response to a certain applied dose is measured at a time $t < t_{eq}$ such that the system has not reached equilibrium, and this is done for all the doses at the same time $t < t_{eq}$, a pre-equilibrium dose-response curve can be plotted (Fig 2B and 2C). In previous articles we studied the temporal evolution of dose-response curves [21,22], finding that several simple signaling components, such as a ligand-receptor system and a covalent modification cycle, shift their dose-response curves in time from right to left. This implies the $EC_{50}$ is a decreasing function of time, a property that could have interesting implications in signaling at a systems level. The **DynR** also evolves in time, as indicated in Fig 2D for a receptor with a single binding site.

We studied how the **DynR** evolves in time for the two models in Fig 1A, finding interesting differences between them. By doing a parameter scan, as explained in the Methods section, we obtained 10000 sets of parameters in the non-identifiability manifold (Fig 1B), solved the corresponding equations, and computed the **DynR** versus time for each one. In this way, we generated two databases, with all the possible **DynR** versus time behaviors for **IB** and **NC** models, respectively. The results are exhibited in Fig 3. By plotting each database, **IB** and **NC**, using a color scale coding for different parameters or combinations of parameters, and undergoing

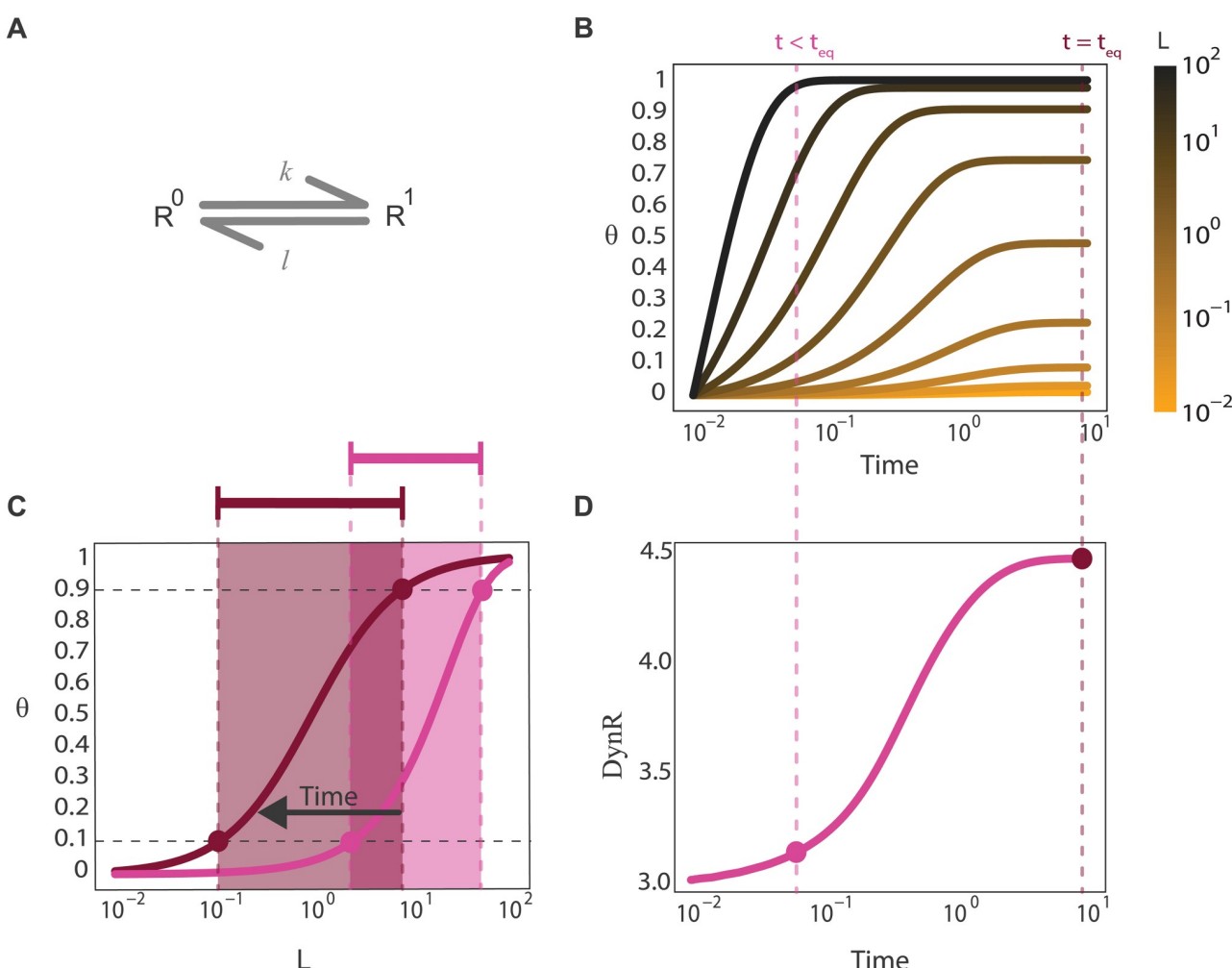

**Fig 2. The dose-response curve evolves in time.** (A). Scheme representing a receptor with a single binding site. (B). Proportion of occupied binding sites ($\theta$) versus time for different doses (L) coded in a color scale. Two vertical dashed lines indicate pre-equilibrium and equilibrium. (C). Proportion of occupied binding sites ($\theta$) versus dose (L) for the two times indicated in (B). $EC_{10}$, $EC_{90}$ and the segment going from one to the other are indicated for the two curves. (D). **DynR** versus time, computed as the log of $EC_{90}/EC_{10}$.

different calculations (see S1 Text), we identified three control parameters as described in the following paragraphs.

For the **NC** model, **DynR** is always an increasing function of time and the **DynR** temporal curves are ordered by $\omega$ (Fig 3B and 3C). For **IB**, on the contrary, **DynR** an be an increasing, decreasing or biphasic function of time and there is no order imposed by $\omega$ (remember that $\omega$ is an effective cooperativity factor in this model) (Fig CA and Fig CC in S1 Text). In Fig 3C we plot **DynR** for very early times, what we call **DynR**($t \rightarrow 0$), versus $\omega$ for both databases. The value **DynR**($t \rightarrow 0$) was estimated analytically (S1 Text) for the **IB** model, finding that it depends on $k_{10}/k_{01}$, as confirmed by Fig 3D and 3F. Since in the **NC** model $k_{10} = k$ and $k_{01} = \omega k$, the ratio $k_{10}/k_{01}$ is $1/\omega$, explaining the results in Fig 3A–3C. Finally, we identified that parameter $l$ (unbinding rate in the **NC** model) controls the time where the inflection point occurs in **DynR** versus time for **NC** model ($t_{ip}$). The calculations in S1 Text explain this last dependency.

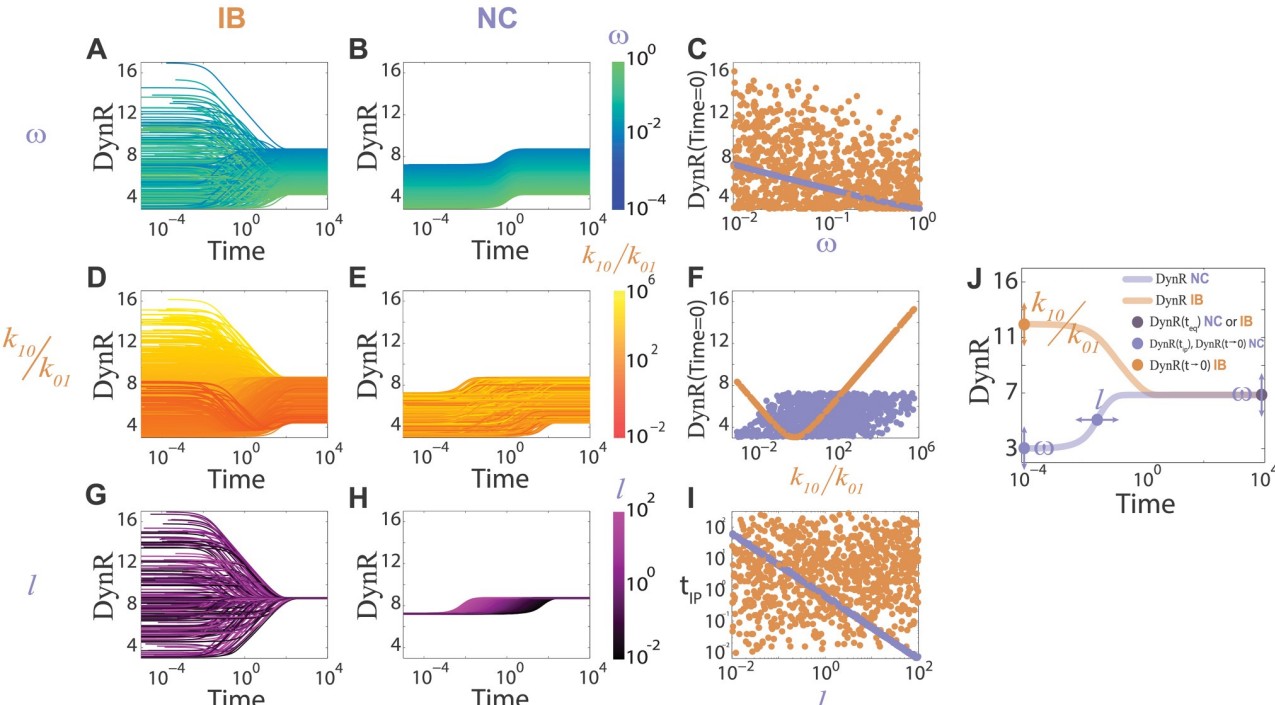

**Fig 3. The dose-response curve evolves in time differently for IB and NC models.** The plots **DynR** versus time (panels (A), (B), (D), (E), (G), (H)) were computed from 10000 parameter sets chosen randomly in the non-identifiability manifold, as explained in the Methods section. For $t > 10^3$ all the curves are in equilibrium, which is the same for both models; for $t < 10^{-3}$ all the curves are in pre-equilibrium. Panels (A), (D), (G) come from the **IB** model; panels (B), (E), (H) come from the **NC** model. In different panels different parameters are coded in a color scale: $\omega$ in (A) and (B), $k_{10}/k_{01}$ in (D) and (E), and $l$ in (G) and (H), we call them control parameters, they control the pre-equilibrium properties of the **DynR** versus time curves. Only in panels (D) and (E), and (C), (F), (I) the complete databases with 10000 sets are included. For clarity, panels (A) and (B) contain only curves with $0.9 < l < 1.1$ (207 curves), and panels (G) and (H) contain only curves with $0.010 < \omega < 0.011$ (218 curves). Complete databases are included in S1 Text (Fig D in S1 Text). (C). **DynR**$(t \to 0)$ versus $\omega$. (F). **DynR**$(t \to 0)$ versus $k_{10}/k_{01}$. (I). $t_{ip}$ (time at which the inflection point in **DynR** versus time occurs) versus $l$. In panels (C), (F), (I), brown dots for **IB**, indigo dots for **NC**. (J). Summary of how each of the considered parameters ($\omega$, $k_{10}/k_{01}$, and $l$) control different properties of the curve **DynR** versus time: $\omega$ controls **DynR**$(t \to 0)$ in **NC** model and **DynR**$(t \to \infty)$ in both models, $k_{10}/k_{01}$ controls **DynR**$(t \to 0)$ in **IB** model, and $l$ controls the time where the curve **DynR** versus time has an inflection point in **NC** model.

In Fig 3J we summarize how each of the control parameters ($\omega$, $k_{10}/k_{01}$, and $l$) control different properties of the curve **DynR** versus time: $\omega$ controls **DynR**$(t \to 0)$ in the **NC** model and **DynR**$(t \to \infty)$ in both models, $k_{10}/k_{01}$ controls **DynR**$(t \to 0)$ in the **IB** model, and $l$ controls the time where the curve **DynR** versus time has an inflection point in the **NC** model. In the following section, we exploit these differences to design an algorithm that allows discrimination between negative cooperativity and ligand binding to independent sites using pre-equilibrium properties of binding curves (*i.e.* the three control parameters $\omega$, $k_{10}/k_{01}$, and $l$).

## An algorithm to discriminate between negative cooperativity and independent binding using pre- equilibrium properties of dose-response curves

**TC algorithm.** As explained in the Introduction, the same question we are addressing in this article, i.e. how to discriminate between negative cooperativity and ligand binding to independent sites, was tackled in a recent paper by Flecha and coworkers [18] following a different approach based also in binding kinetics. Their approach implies fitting the experimental time courses with the two models under consideration. The model that better describes the data is selected by statistical criteria. Based on these ideas and adding constrains to the fitting given

the macroscopic equilibrium constants obtained from the data, we developed the following algorithm (TC algorithm, TC from "time course"). The input is θ vs. time for different L (Fig 4A, data in this example was generated with the **IB** model), and the output is the guessed identity of the data, i.e., whether it belongs to the **IB** or **NC** model. The TC algorithm has three steps:

**Step 1**. Fit the curve θ vs. L at equilibrium with Eq (2) to get $(K, \omega)$ for the **NC** model and $(K_{10}, K_{01})$ for the **IB** model (Fig 4B).

**Step 2**. Simulate the **IB** and **NC** models (Fig 1A and Eqs (9) and (10)) with the constraints obtained in Step 1. The **NC** model has three parameters $(k, l, \omega)$ and the **IB** model has four parameters $(k_{10}, k_{01}, l_{10}, l_{01})$. In Step 1, two fitted values are obtained in each model, resulting in only one free parameter for **NC** ($k$ and $l$ with the constraint $l/k = K$) and two for **IB** ($k_{10}, k_{01}, l_{10}, l_{01}$ with the constraints $l_{10}/k_{10} = K_{10}$ and $l_{01}/k_{01} = K_{01}$). We use the optimization routine *fminsearch* in Matlab to fit the input data with these two models and its free parameters (Fig 4C).

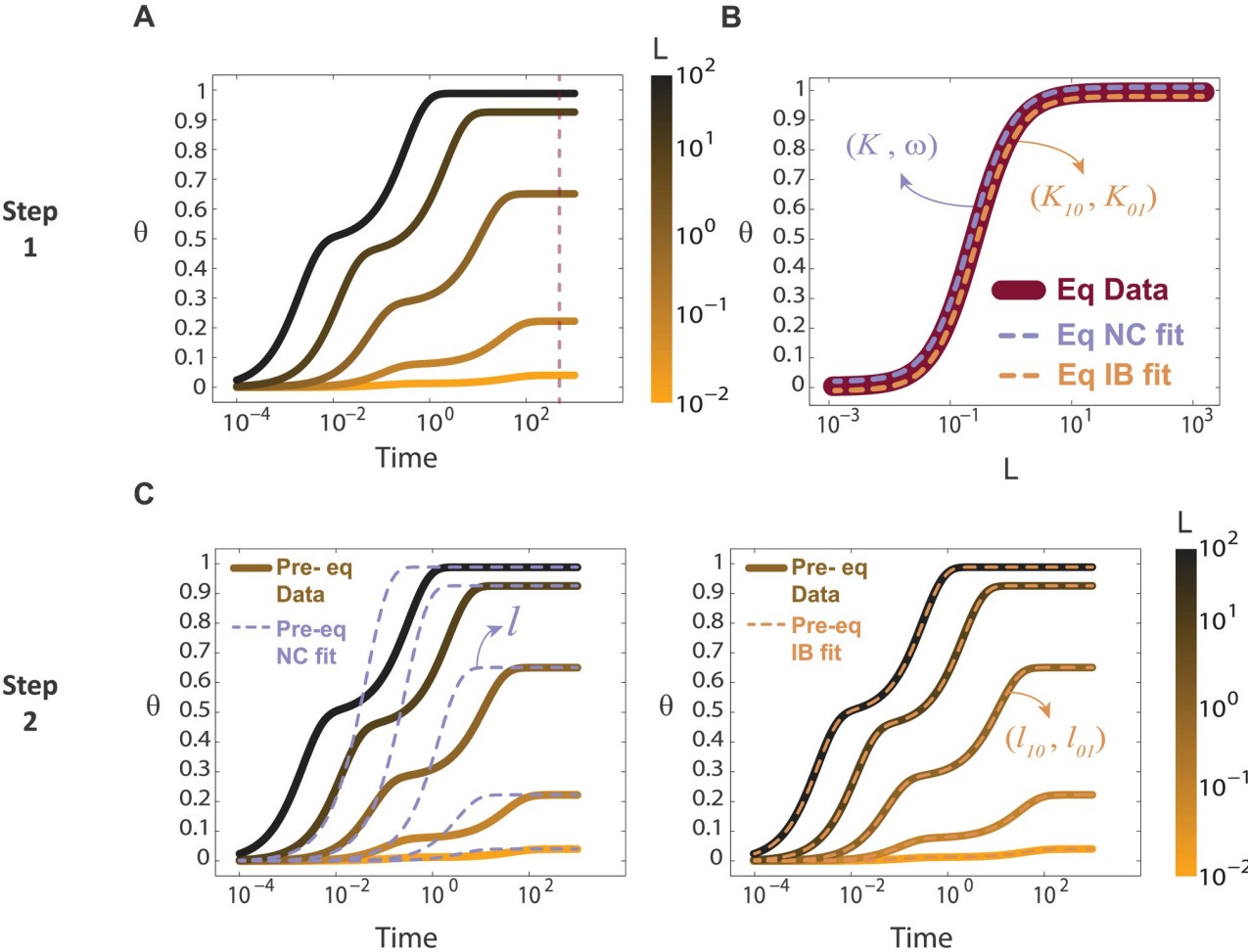

**Fig 4. TC algorithm.** (A). Input: **θ** vs. time for different L, coded in a colorscale. Dashed line indicate equilibrium. (B). **θ** vs. L at equilibrium fitted with Eq (2) to get ω, K and $K_{10}$, $K_{01}$. (C). Time courses simulated with **NC** model (left) and **IB** model (right) with the constraints obtained in (B). Only the best fit (dashed lines) is included in each plot. For the example in this figure the data was generated with the **IB** model (parameters in Table A in S1 Text).

**Step 3**. We compute the L2 norm of the difference between the input data and the two two-variables functions $\theta$(L, t) obtained with the fitting, one for each model. The fact that the **IB** model has one extra parameter compared with the **NC** model introduces a penalization in the fitting (since the higher the number of parameters, the easier to fit a given target), resulting in a corrected comparison between **IB** and **NC** distances (see S1 Text). The lower of the two distances (**IB** and **NC**) provides the output of the procedure. Step 3 is illustrated in the upper row of Fig 5.

## Improving the TC algorithm by using DynR information: TC+DR algorithm

Having the two distances from Step 3 in the TC algorithm, we define a quantity **C** from an F-test [24], using Matlab *fcdf* (F cumulative distribution function) which represents the probability of each dataset to belong to **NC** or **IB**, considering that **IB** has one extra parameter (details in the S1 Text, Fig FA and Fig FB in S1 Text, where the TC+DR algorithm efficiency dependence with **C** is evaluated). If the minimum distance in **IB** fit is much lower than the one in **NC**, **C** is approximately 1. On the other hand, if the minimum distance in **NC** fit is much lower than the one in **IB**, **C** is approximately 0. Oppositely, if the two minimum distances are similar, this would mean that is not possible to decide which model generated the data, in that case **C** is approximately ½. Summarizing, **C** is a quantity between 0 and 1, ½ meaning absolute uncertainty and 0 or 1 meaning absolute certainty to **NC** or **IB**, respectively. By fixing two threshold values of **C**, we set the criteria on how much different these two distances must be to decide which model originated the data. The closer to ½ this value is, makes the discrimination less strict, i.e. it requires a lower difference between the two distances. We fix the arbitrary thresholds in 1/3 and 2/3: if **C** is in between them we assume that the information provided by the TC algorithm is not enough to decide which model generated the data and continue with the following steps based on the information described in Fig 3. We compute the curve **DynR** versus time from the data, we call it the **target curve** (Fig 5, second row), and focus on the three control parameters analyzed in Fig 3 and obtained from Steps 1 and 2: $\omega$, $k_{10}/k_{01}$ and *l*. In what follows we describe different sequential tests or checkpoints that are applied to a dataset in order to decide the identity of the underlying model (**NC** or **IB**).

**Checkpoint 1: target curve global behavior**. If the target curve is not monotonically increasing, i.e. it is biphasic or monotonically decreasing, then it is concluded that the data comes from an <u>IB model</u> (see databases in Fig 3). If it is monotonically increasing, then we continue by quantifying several aspects of the target curve.

**Checkpoint 2: analyzing DynR**(t → 0). We get the value of **DynR** at the earliest time available. If the shape of the target curve allows assuming that the value of **DynR** at the earliest time is a reasonable estimation of **DynR**(t → 0) (see S1 Text), the question is if it is well described by the function **DynR**(t → 0) vs. $\omega$ (Fig 3C), by the function **DynR**(t → 0) vs. $k_{10}/k_{01}$ (Fig 3F), or by both. In this last case, we go to the next checkpoint. The functions **DynR**(t → 0) vs. $\omega$ and **DynR**(t → 0) vs. $k_{10}/k_{01}$ are numerically obtained by the databases analysis described in Fig 3. Having estimations of $\omega$ from the equilibrium fit in Step 1 and $k_{10}/k_{01}$ from Step 3, we compare the value of **DynR**(t → 0) of the target curve with **DynR**$_{NC}$(t → 0) at the estimated $\omega$ and **DynR**$_{IB}$(t → 0) at the estimated $k_{10}/k_{01}$. If the difference between **DynR**(t → 0) and **DynR**$_{NC}$(t → 0) at the estimated $\omega$ is higher than an arbitrary threshold (see S1 Text), then it is concluded than the data comes from an <u>IB model</u>. If it is lower, we analyze the difference between **DynR**(t → 0) and **DynR**$_{IB}$(t → 0) at the estimated $k_{10}/k_{01}$, if it is higher than an arbitrary threshold (see S1 Text), it its concluded than the data comes from an <u>NC model</u>. If

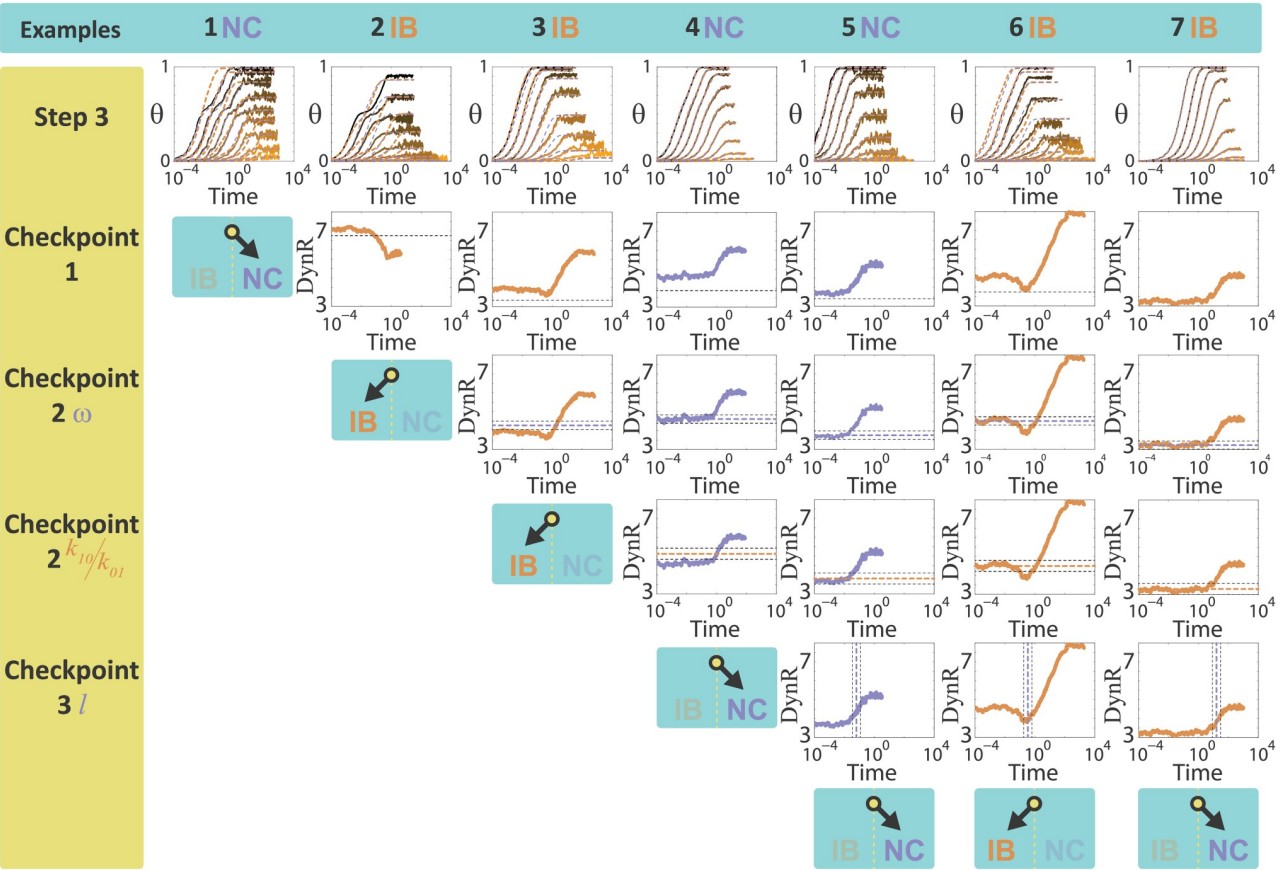

**Fig 5. A combined algorithm to distinguish between IB and NC models based on the differences in binding kinetics and DynR temporal evolution.** Columns contain 7 examples of data generated with the two models. The identity of each example (**IB** or **NC**) is indicated in each column header and follows a color code, brown for **IB** and indigo for **NC**. Parameters that generated the data are listed in Table A in S1 Text. The data was produced using stochastic simulations as explained in the Methods section, with a total number of receptors of 1000. Data appears to be noisier for longer times due to the logarithmic scale used in the figure. **First row**. Input data ($\theta$ vs. time for different values of L) together with the best **IB** fit (dashed brown lines) and the best **NC** fit (dashed indigo lines). **Second row**. DynR versus time target curve. **Third row**. Predicted $\mathbf{DynR}(t \rightarrow 0)$ according to the value of $\omega$ obtained from Step 1, indicated with an indigo horizontal dashed line. The upper and lower lines are the corresponding thresholds (see S1 Text). **Fourth row**. Predicted $\mathbf{DynR}(t \rightarrow 0)$ according to the value of $k_{10}/k_{01}$ obtained from Step 2, indicated with a brown horizontal dashed line. The upper and lower lines are the corresponding thresholds (see S1 Text). **Fifth row**. Predicted time for the $\mathbf{DynR}(t)$ inflection point ($t_{ip}$), according to the value of $l$ obtained from Step 2, indicated with an indigo vertical dashed line. The dashed lines to the left and to the right are the corresponding thresholds (see S1 Text). In **Example 1**, $C$ is higher than the threshold, so the algorithm indicates the data comes from an **NC** model. In Examples 2 to 7, the fits are both bad (2, 3, 6) or both good (4, 5, 7), so $C$ does not allow a decision and further analysis is needed. In **Example 2**, DynR is decreasing with time, this can only happen in the **IB** model. In Examples 3 to 7, DynR is an increasing function of time. In **Example 3**, the range predicted by the **NC** model for $\mathbf{DynR}(t \rightarrow 0)$ does not contain the value of $\mathbf{DynR}(t \rightarrow 0)$ from the target curve, so the data comes from an **IB** model. In Examples 4 to 7, the value of $\mathbf{DynR}(t \rightarrow 0)$ from the target curve is within the predicted range, so both models can explain the data and further analysis is needed. In **Example 4**, the range predicted by the **IB** model for $\mathbf{DynR}(t \rightarrow 0)$ does not contain the value of $\mathbf{DynR}(t \rightarrow 0)$ from the target curve, so the data comes from an **NC** model. In Examples 5 to 7, the value of $\mathbf{DynR}(t \rightarrow 0)$ from the target curve is within the predicted range, so both models can explain the data and one last checkpoint is needed. In **Examples 5 and 7**, the time where the inflection point in the **DynR** curve occurs ($t_{ip}$) is well described by the range predicted by the **NC** model, so it is concluded that these two examples correspond to the **NC** model. In Example 5 this conclusion is right, in Example 7 is not. This Example illustrates the only group of situations where the algorithm fails. A global analysis indicates that only the 5% of the cases fall in this group (this number corresponds to $R_0 = 1000$, which is the value used in the seven examples analyzed in this figure). Finally, in Example 6, $t_{ip}$ is outside the range predicted by the **NC** model, so it is concluded that the data comes from the **IB** model.

$\mathbf{DynR}(t \rightarrow 0)$ if well described by both $\mathbf{DynR_{NC}}(t \rightarrow 0)$ and $\mathbf{DynR_{IB}}(t \rightarrow 0)$, then the next checkpoint is needed.

**Checkpoint 3: analyzing $\mathbf{t_{ip}}$.** We end up here in cases where $C$ is lower than the threshold (i.e. the fitting obtained by steps 1–3 are both good or both bad), the target curve **DynR** is an

increasing function of time, and $\mathbf{DynR}(t \rightarrow 0)$ is well described by both $\mathbf{DynR_{NC}}(t \rightarrow 0)$ and $\mathbf{DynR_{IB}}(t \rightarrow 0)$, meaning that up to this point both models could explain the data. Having the estimation of $l$ from Step 3, we compare the $t_{ip}$ **Data** (time of the inflection point of the target curve) with the $t_{ip}$ NC at the estimated $l$, according to the function in Fig 3I. If the difference between $t_{ip}$ **Data** and $t_{ip}$ NC is higher than an arbitrary threshold (see S1 Text), then it is concluded than the data comes from an __IB__ model. If it is lower, it is concluded that the data comes from the __NC model__. This last conclusion is mistaken only in 26, 7, 5, and 1.5% of cases, each value corresponding to different total number of receptors ($R_0$ = 10, 100, 1000 and $\infty$ respectively), i.e. to different levels of stochasticity.

The procedure described by the three checkpoints is illustrated in Fig 5, where 7 examples were selected indicating the different possible outcomes of each checkpoint.

## Performance of TC and TC-DR algorithms

In order to test and compare the performance of the two algorithms (TC alone and TC+DR), we computationally simulated **IB** and **NC** models both with deterministic and stochastic methods, in this last case for different total number of receptors ($R_0$ = 10, 100, 1000) (see Methods for details). Examples of dose-response curves for different times and the corresponding **DynR** versus time curves, obtained with $R_0$ = 10 and $R_0$ = 1000, are included in Fig 6A. We then randomly selected 100 different sets of parameters for each model, chosen by Latin Hypercube Sampling [25], and applied both algorithms to all of them (Fig 6B). The combined one has very high performance (75% of correct definitions) even for very noisy simulations ($R_0$ = 10).

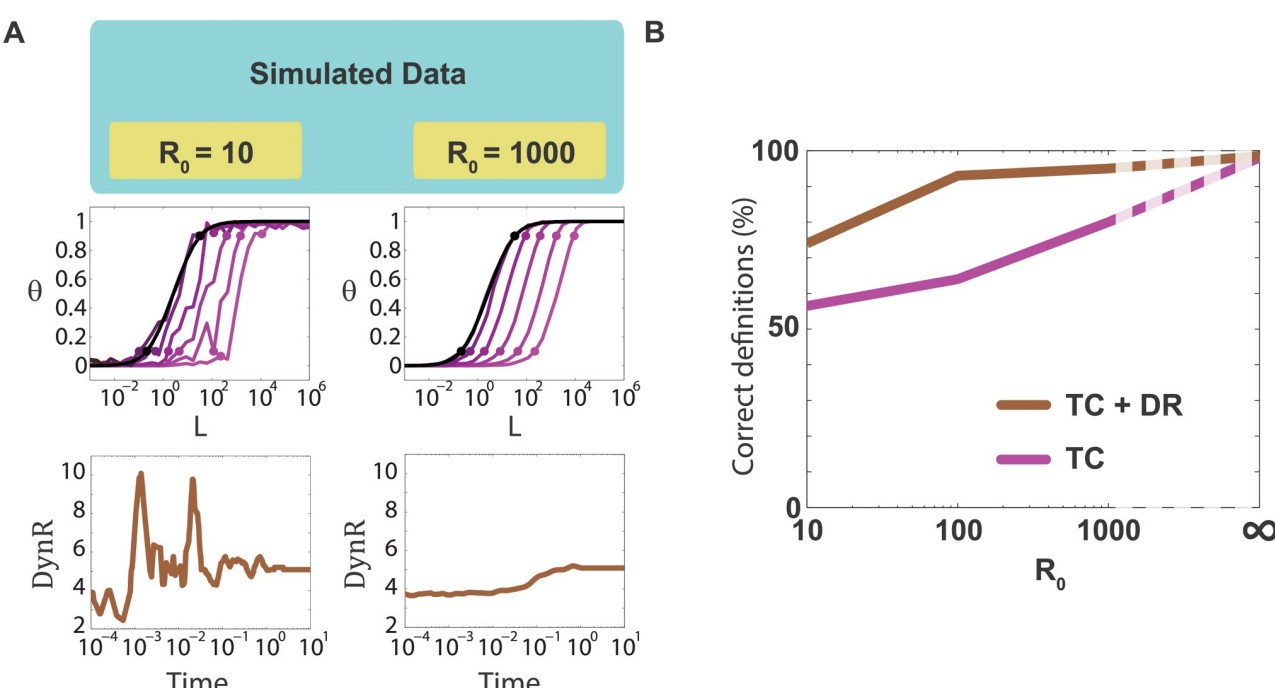

**Fig 6.** (A). **Testing the algorithms with simulated stochastic data**. Dose-response curves for different times in color scale (time evolves from pink to black). Examples for $R_0$ = 10 (**up and left**) and $R_0$ = 1000 (**up and right**) ($R_0$ is the number of total receptors). Simulated **DynR** versus time (target curves), for $R_0$ = 10 (**bottom and left**) and $R_0$ = 1000 (**bottom and right**). Parameters provided in Table A in S1 Text. (B). **Performance of TC and a combined TC and DR algorithms in discriminating between IB and NC models**. Performance of the TC algorithm alone (pink line) and combined with DR analysis (green line), in terms of the total number of receptors, where corresponds to deterministic simulations.

## The algorithm applied to experimental data

Even though we tested the algorithms with noisy simulated data, there are other sources of uncertainty when applying the different steps and checkpoints to experimental data, namely: the temporal resolution at which the data was acquired, the step in ligand concentrations at which the dose-response curve was obtained, how well **DynR**(t → 0) can be estimated, and if the equilibrium is reached or not. To test all this together, we selected four sets of experimental data from the literature.

Before describing the data, we consider all the possible scenarios with two binding sites. These two binding sites can be identical or different. Each category can have cooperativity or not. So, the two binding sites space is divided in four, as summarized in Fig 7. The identical and cooperative region contains the **NC** cases studied in this paper (and positive cooperativity cases as well) (Fig 7 up and right), while the different and non-cooperative region contains the **IB** cases studied in this paper (Fig 7 bottom and left). The identical and non-cooperative region behaves as a single binding site model, so it is well described by both **NC** and **IB** models, provided that the cooperativity factor is ω = 1 or $K_{10} = K_{01}$, respectively (Fig 7 up and left). Summarizing, having data that comes from a two binding site experiment, and applying the ideas developed in this paper, the possible outcomes are: **NC** describes the data, **IB** describes the data, both **NC** and **IB** describe the data with ω = 1, none of them describe the data, covering all possibilities with two-binding sites. In Fig 7 we include the output of studying experimental data sets corresponding to these four cases (more details in the S1 Text).

The **NC** experimental data in Fig 7 (upper right panel) comes from an article studying ligand binding to DNA [26] (data extracted from Fig 3D of the cited article). The ligand is a chiral helical macrocyclic lanthanide complex, and it binds to a GC-duplex DNA sequence, which is a periodic sequence with 17 GC repetitions. The binding sites are identical because the ligand binds GC. The data was acquired with Fourier transform-surface plasmon resonance (FT-SPR) experiments [27]. In the article it is recognized that no positive cooperativity emerges from the data. We found that an **NC** model explains the data with parameters in agreement with those reported in the article (see Table 1). This is consistent with the fact that the ligand molecules size is similar to the receptor size (the 17 GC repetitions), leaving less space for a second binding.

The **IB** experimental data (Fig 7, lower left panel) comes from an article performing a kinetic analysis of ligand binding to interleukin-2 receptor complexes created on an optical biosensor surface [28] (data extracted from Fig 1C of the cited article). The interleukin-2 receptor (IL-2R) is composed of at least three cell surface subunits, IL-2Rα, IL-2Rβ, and IL-2Rγ$_c$. On activated T-cells, the α- and β-subunits exist as a preformed heterodimer that simultaneously captures the IL-2 ligand as the initial event in formation of the signaling complex. The data analyzed here comes from an experiment in which they compare the binding of IL-2 to biosensor surfaces containing either the α-subunit, the β-subunit, or both subunits together. Equilibrium analysis of the binding data established IL-2 dissociation constants for the individual α- and β-subunits of 37 and 480 nM, respectively. Surfaces with both subunits immobilized, however, contained a receptor site of much higher affinity, suggesting the ligand was bound in a ternary complex with the α- and β–subunits. Since the experiment we selected was done with an excess of the β–subunit in a 1:3.4 molar ratio, this data is expected to behave as **IB**, one binding site being the β–subunit, the other one being the preformed heterodimer. The data was acquired with SPR experiments.

The data corresponding to identical binding sites with no cooperativity (Fig 7, upper left panel) comes from an article measuring the binding between carbonic anhydrase isozyme II (the ligand) and carboxybenzenesulfonamide (the receptor) [27] (data extracted from Fig 6A

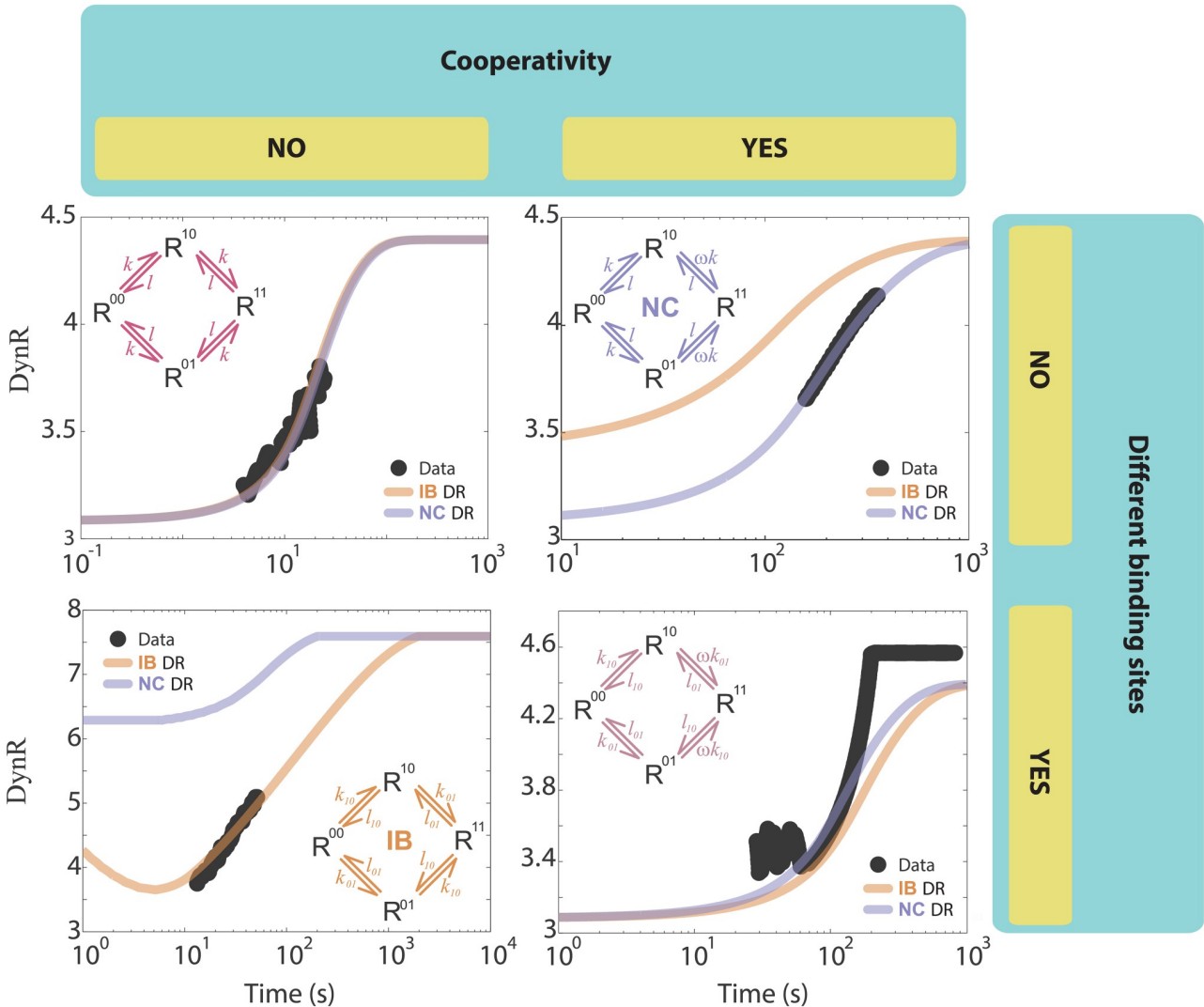

**Fig 7. TC + DR algorithm applied to experimental data.** The two binding sites space is divided in four: left/right panels do not have/have cooperativity, upper/lower panels do not have/have different binding sites. In all four panels **DynR** vs. time is plotted, filled dots are the data, indigo/brown solid line is the **DynR** expected from the **NC/IB** model. Schemes with the microscopic models are superimposed in each panel. **Up and left**, data from a one site model, both curves (**DynR** from **NC** and from **IB**) are the same and fit the data. **Up and right**, **NC** experimental data, **bottom and left**, **IB** experimental data (details of the steps and checkpoints applied to these two datasets are in Fig G in S1 Text). **Bottom and right**, data from a two different sites binding experiment with positive cooperativity, neither **NC** nor **IB** fit the data, as expected. In the **NC** example, time goes from 0 to 700 s with a step of 0.1; ligand goes from 0.01 to 3 μM, with a non-uniform step totalizing 13 doses (0.01, 0.05, 0.075, 0.1, 0.15, 0.2, 0.3, 0.5, 1, 1.5, 2, 2.5, 3 μM). In the **IB** example, time goes from 0 to 500 s with a step of 0.1; ligand goes from 3.9 to 2000 nM, each dose is the double of the previous one totalizing 10 doses. In the up and left example, time goes from 0 to 40 s with a step of 0.025; ligand goes from 0.1 to 25.6 μM, each dose is the double of the previous one, totalizing 9 doses. In the bottom and right example, time goes from 0 to 250 s with a step of 0.025; ligand goes from 0.7 to 200 nM, each dose is approximately 1.85 times the previous one, totalizing 10 doses. Parameters obtained from the fits and compared with those in the articles where the data comes from are included in Table 1.

of the cited article). The data was acquired with SPR experiments and reported to correspond to a single-site binding model. Finally, the data corresponding to different and cooperative binding sites (Fig 7, lower right panel) comes from an article studying the glucocorticoid receptor binding to genomic response elements [29] (data extracted from Fig 5A, upper left panel, of the cited article). The response elements were synthetized in the analyzed article and are different (Pal-R is the one corresponding to the data extracted), thus leading to different

**Table 1. Equilibrium ratios for the experimental databases.** For each of the four experimental databases considered (listed on the first column: Equal Sites, **NC**, **IB**, Diff & Coop) the equilibrium dose-response curve was fitted both with the mathematical expression corresponding to an **IB** description and with that of an **NC** description (Eqs (1) and (2) in the main text), obtaining $K_{10}$ and $K_{01}$, or K and K/ω, respectively. For the experimental databases labeled Equal Sites, **NC** and **IB** both fittings are good, because of the indistinguishability reasons. The considered articles fitted the data (dose-response curve in equilibrium) using a two-site model (**TS**); this model is described in detail in the S1 Text. The **TS** model provides two parameters, $K_1$ and $K_2$. By comparing the equilibrium dose-response curves from **IB** and **NC** models, with that of the **TS** model, a mathematical relationship between $K_1$-$K_2$ and $K_{10}$-$K_{01}$, and $K_1$-$K_2$ and K-K/ω was obtained. Summarizing, the papers provide $K_1$-$K_2$, and using the mentioned relationships the inferred $K_{10}$-$K_{01}$ and K-K/ω are calculated from those reported values, to be compared with the inferred $K_{10}$-$K_{01}$ and K-K/ω from fitting with **IB** and **NC** models. Data coming from Diff and Coop does not have reported $K_1$ nd $K_2$, that is why that row is empty.

| Experimental dataset | Equilibrium Ratios | | | | | | | | |
|---|---|---|---|---|---|---|---|---|---|
| | $K_{10}$ | | $K_{01}$ | | K | | K/ω | | |
| | fitting with IB | reported values | fitting with IB | reported values | fitting with NC | reported values | fitting with NC | reported values | unit |
| Equal Sites | 0.81 | 0.92 | 0.81 | 0.92 | 0.99 | 0.92 | 0.67 | 0.92 | µM |
| NC | 1.0 | 1.0 | 540 | 419 | 8.0 | 2.0 | 270 | 210 | nM |
| IB | 0.38 | 0.76 | 3.8 | 9.2 | 0.59 | 1.42 | 7.0 | 5.0 | µM |
| Diff & Coop | 4.2 | | 4.2 | | 5.7 | | 3.2 | | nM |

binding sites; reported Hill coefficients higher than 1 support positive cooperativity in this binding experiment. This data was also acquired with SPR experiments.

The visual inspection indicates that **NC**, **IB** and the data corresponding to identical binding sites with no cooperativity are fitted with accuracy. The parameters obtained in each case are in agreement with those reported in the corresponding papers, as summarized in Table 1 (more details in S1 Text).

Table 1 shows the equilibrium ratios obtained for the four experimental data cases described in Fig 7, fitting the equilibrium dose-response curve from each one with both model's predictions (Eq (1) for $K_{10}$ and $K_{01}$ and Eq (2) for K and ω). These ratios are compared with those obtained from data reported in the mentioned papers.

## Discussion

This article deals with a problem that is within the broad problematic of model discrimination in systems biology [30], meaning that structurally different computational models fit a set of experimental data equally well, resulting in more than one molecular mechanism being able to explain available data. This area of research is certainly enriched by adding dynamics into the methodology to address model ambiguity, either by fitting not only steady-states but also time courses [18] or by designing dynamic stimuli that, in stimulus–response experiments, distinguish among parameterized models with different topologies [31]. The models that are the center of the present article only fit steady-state data equally well, but not kinetic data, which is the key in discriminating between both mechanisms. This last statement places our approach close in essence to those that apply time varying inputs to distinguish closely related models of biochemical systems. We hypothesize that, because of those similar bases, the method proposed in this article may be used to infer model structure from a set of plausible candidates in different biochemical models.

Previously we have shown that if the input-output curve of a ligand-receptor reaction is measured before it reaches equilibrium, it is shifted to the right of its equilibrium position and it moves from right to left as time evolves. More specifically, the $EC_{50}$ decreases over time. We labelled systems with this property as shifters. As the $EC_{50}$ evolves in time, so do the $EC_{10}$ and the $EC_{90}$, the two of them define the dynamic range of the reaction (**DynR** = $\ln(EC_{90}/EC_{10})$) which is in turn a function of time. In this article, we studied the **DynR** temporal evolution for receptors having two binding sites, which are also shifters, with extensions to other successive binding events not involving receptors. The focus of this work was on the indistinguishability

problem between negatively cooperative identical sites and independent and different sites: both scenarios lead to the same equilibrium binding curve. Different researchers have studied the kinetics leading to that equilibrium binding curve, finding that **NC** and **IB** follows different kinetics and this is, then, a way to distinguish them. However, when data is noisy, the kinetic discrimination based on fitting binding time courses is challenging.

The method we propose in this article to tackle the **NC—IB** indistinguishability problem takes a global measurement over the input-output curve, and repeats this procedure as time evolves, obtaining de **DynR** versus time curve, or the target curve as we called it in this article. The more important feature that makes the method robust and successful is that it relies on the estimation of $\omega$ (the cooperativity factor, Fig 1A), done over the equilibrium input-output curve, together with the fact that the target curves are ordered by $\omega$ in the **NC** model, as shown in Fig 3B. This feature alone (called Checkpoint 2 in the paper) can solve the indistinguishability problem in 88% of the cases. The target curves in **NC** model are always increasing, their initial and final values are functions of $\omega$, and the time corresponding to the inflection point of the curve is a function of $l$, the unbinding rate. All these characteristics make the target curve a good observable to analyze in the context of deciding if the data comes from **NC** or **IB** models.

Why is it exactly that the proposed approach to distinguish between **IB** and **NC** scenarios resulted in an improvement from previously implemented methodology [18], given that both of them use kinetic information?. As explained, our methodology uses both steady-state and kinetic information, and both local and global information (local and global with respect to the dose-response curve, distinguishing if it uses data from one or a few doses or from the complete curve). What makes the model so efficient is that it deals with kinetic data but not that one associated to the time evolution of different variables, instead, it deals with **DynR**(t). In a binding process, the fitting of the variables' time courses depends on the binding and unbinding rates and on the amount of ligand. However, **DynR**(t $\rightarrow$ 0) depends on the ratio between those rates, which leads to $\omega$ in the **NC** scenario and to $k_{10}/k_{01}$ in the **IB** one, and does not depend on the ligand concentration. The dependency with an unique parameter that is estimated from the equilibrium dose-response curve resulted in a clear advantage.

The approach proposed in this article seems to be robust under noisy data and other sources of uncertainty and is based on the preliminary knowledge that there are only two binding sites and relies on having sufficient temporal and dose resolution, that **DynR**(t $\rightarrow$ 0) can be well estimated and that saturation in the dose-response curve is reached. Some other implicit assumptions are the following. First, it compares pure **IB** and **NC** mechanisms, heterogeneous binding is not covered by the approach. Second, ligand depletion was not considered in the present form of the method, its inclusion is part of current efforts to improve the approach. Third, the method is based on the proportion of occupied binding sites ($\theta$). If, instead, what is measured is the receptor with two bound ligands ($R^{11}$), the method cannot be applied in its current form, modifications in this direction are also part of current research. Fourth, the bindings that are covered by the method are such that they are well described by the law of mass action.

The method was based on the properties of the target curve as a function of different parameters, these properties came to light both by numerical exploration (Fig 3) and by analytical calculations (details in the S1 Text). Importantly, the method was tested with simulated data and with experimental data as well. The numerical data was generated considering the noise introduced by chemical reactions. As expected, the performance of the method decreases as molecular noise increases (Fig 6), but in the worst scenario tested, the performance decreased no further than 75%, meaning that in 75% of the cases the algorithm made a correct guess.

In the case of experimental data, for some of the datasets we used, the microscopic model was known, and our method confirmed that information. In the case of the **NC** dataset, even when it is not stated that the binding scheme corresponded to negative cooperativity, our results support that explanation (Fig 7). Not only our method is successful in identifying the microscopic model, but also in obtaining its parameters (Table 1).

The method presented in this article relies on different thresholds, as studied in detail in Fig E and Fig F in S1 Text. Fig F in S1 Text evaluated the accuracy of the predictions as a function of threshold **C** (Fig FB in S1 Text). Regarding the other thresholds involved in the method (Fig E in S1 Text) we did not include results with variations of their values, this effect can be seen from the mentioned figure. Any change in the selected value of those thresholds will result in a worsening of the outcome. The reason for this is that those values were obtained from an optimization procedure (that was performed with simulated data with a total number of receptors of 1000). There is still something that could be done for each experimental database in which the method is applied: from the noise in the temporal curves it is possible to infer the number of receptors, then, the thresholds for that database can be reobtained by using simulated data with that number of receptors. In this way, one can obtained the best set of thresholds for a given database.

It is unusual to find a paper focusing on cooperativity, as this one, and not dealing with the Hill coefficient. Instead of that, and based in our previous papers [21,22], we focused on the dynamic range as a function of time. One of the definitions of the Hill coefficient ($n_H$) is related to the **DynR** in this way: $n_H = \ln(81)/\ln(EC_{90}/EC_{10}) = \ln(81)/$**DynR**. However, we decided to focus on **DynR** because $n_H$ is usually associated with the slope of the curve and, in the problem we are tackling, a single slope is not always possible to define (Fig 1C).

If several binding sites are suspected and the **DynR** of the equilibrium input-output curve is higher that what is expected for a single binding site (**DynR** = 1.9), then the method we proposed helps unveiling the microscopic details behind the data. Regarding possible extensions of the method and the analysis in this article, even though it is possible to rebuild the procedure for 3 or more binding sites, the complexity is high since different combinations arise (taking 3 binding sites as an example, the 3 of them could be independent, 2 cooperative and 1 independent, 2 independent and 1 cooperative, all of them cooperative). Extension to several binding events not involving ligand-receptor reactions is presented related to experimental datasets **NC** and **IB** (Fig 7).

In order to know if the approach presented in this article can be used to characterize a particular model and identify features that contribute in distinguishing it from other models with similar or identical dose-response curves, it is important to study the function **DynR(t)** in those models. In a previous article [22] we have characterized which signaling systems display time-evolving $EC_{50}$ (t) and **DynR(t)**, finding that nearly all biochemical processes operate in this way and that this mechanism may be ubiquitous in cell signaling systems. Therefore, studying and characterizing **DynR(t)** is a promising tool for characterizing every signaling model based on biochemical processes. If, in addition, an indistinguishability problem involves that particular model under consideration as is the case for **IB** and **NC** models, or for adaptation models in which the adaptive response could result from mainly two different types of models [32], then the study of the function **DynR(t)** could help solve the problem, as was shown in detail in this article. In this direction, extensions of the essence of the method, i.e. different global behaviors of **DynR** versus time as a way to distinguish different but close scenarios, are under study in our lab and have promising results in models for adaptation in signaling (some preliminary results in [21]). The proposed approach is useful to distinguish between some candidate models that fit the data equally well, but not to infer model structure. However, if different biochemical models are grouped in classes according to a particular

criteria related to its structure, and the function **DynR(t)** shares common properties within a class, the study of the target function could help to infer model structure.

## Methods

All the algorithms used in this paper were written in Matlab R2014a.

### 1. Dose-response curves and dynamic range

In this article we consider dose-response curves, in which the <u>dose</u> is the amount of ligand L, and the <u>response</u> is the proportion of occupied binding sites:

$$\theta = \frac{R^{10} + R^{01} + R^{11}}{2R_0} \tag{7}$$

where $R^{10}$ and $R^{01}$ are the configurations with only one site occupied, $R^{11}$ represents the receptor with two bound ligands, and $R_0$ is the total amount of receptors.

The dose-response curves were characterized by their <u>logarithmic dynamic range</u> (**DynR**), defined as:

$$\mathbf{DynR} = \ln\left(^{EC_{90}}/_{EC_{10}}\right) \tag{8}$$

where $EC_{10}$ ($EC_{90}$) is the concentration of ligand that gives 10% (90%) of the total occupation of sites. **DynR** is the range of inputs or doses for which the system can generate distinguishable outputs or responses.

### 2. Deterministic numerical simulations

We described both models (**IB** and **NC**) using mass action law, where $R_0$ is the total receptor concentration. We computationally integrated these two systems of equations for each given set of parameters, for 1000 different values of ligand concentration each one between $10^{-3}$ and $10^8$, in uniform $\log_{10}$ scale:

$$\frac{d}{dt}\begin{pmatrix} R^{10} \\ R^{01} \\ R^{11} \end{pmatrix} = \begin{pmatrix} -(l_{10} + k_{10}L + k_{01}L) & -k_{10}L & l_{01} - k_{10}L \\ -k_{01}L & -(l_{01} + k_{10}L + k_{01}L) & l_{10} - k_{01}L \\ k_{01}L & k_{10}L & -(l_{10} + l_{01}) \end{pmatrix} \begin{pmatrix} R^{10} \\ R^{01} \\ R^{11} \end{pmatrix} + R_0\begin{pmatrix} k_{10}L \\ k_{01}L \\ 0 \end{pmatrix} \tag{9}$$

$$\frac{d}{dt}\begin{pmatrix} R^{10} \\ R^{01} \\ R^{11} \end{pmatrix} = \begin{pmatrix} -(l + kL + \omega kL) & -kL & l - kL \\ -kL & -(l + kL + \omega kL) & l - kL \\ \omega kL & \omega kL & -2l) \end{pmatrix} \begin{pmatrix} R^{10} \\ R^{01} \\ R^{11} \end{pmatrix} + R_0\begin{pmatrix} kL \\ kL \\ 0 \end{pmatrix} \tag{10}$$

From the integrated variables $R^{ii}(L, t)$, we obtained $\theta(L, t)$ using Eq (7) and then $EC_{10}(t)$ and $EC_{90}(t)$ numerically and **DynR(t)** from Eq (8). These 10000 different sets of **DynR(t)** curves are the ones shown in Fig 3 and Fig A in S1 Text, with different parameters in different colored scales and with different filters.

### 3. Random parameter scan

We sampled parameters $k$, $l$, $l_{10}$, $l_{01}$ between $10^{-2}$ and $10^2$ and $\omega$ between $10^{-2}$ and 1, using Latin Hypercube Sampling [25], for 10000 different sets. The missing parameters, $k_{10}$ and $k_{01}$, where obtained from the non-identifiability conditions (Eqs (3) and (4)) to make sure we were comparing pairs of sets (one of **IB** and the other of **NC**) where the problem exists, this means,

in the non-identifiable manifold (Fig 1B). Thus, we are considering 4 orders of magnitude in uniform $\log_{10}$ scale. 3 of the parameters have dimensions of inverse of concentration*time ($k$, $k_{10}$ and $k_{01}$) and other 3 have dimensions of inverse of time ($l$, $l_{10}$ and $l_{01}$). $\omega$ has no dimensions. The interpretation of the results depends yet on the choice of the reference unit concentration (the '0' in log scale). For example, if the reference dimensional concentration is chosen as 0.1 μM, this leads to interpreting the scanned intervals as being in the range [1 nM, 10 μM], which seems reasonable as intracellular concentrations [33]. However, this is just an example and the choice of the reference unit concentration remains a degree of freedom in our numerical methodology.

## 4. Stochastic numerical simulations

To obtain stochastic simulated data to test our protocol, we used Gillespie's algorithm [34] that considers the probability of each reaction to occur. This method needs the total number of receptors, ligands and the volume where the reactions occur. We took 100 sets of parameters for each model, chosen as described in the previous section, and repeated this for 4 different total number of receptors ($R_0$), 10, 100, 1000 and infinite, this last case representing deterministic simulations and thus integrated by mass action law (Eqs (9) and (10)), as before. The volume was chosen to result in the same ligand concentration in all cases, which were 20 different concentrations between $10^{-3}$ and $10^7$ in uniform $\log_{10}$ scale, to make sure to reach the 90% of occupied sites for any set of parameters and represent a plausible experiment. As for the deterministic simulations, we obtained $\mathbf{\theta}(L, t)$ using Eq (7) and then $EC_{10}(t)$ and $EC_{90}(t)$ numerically and $\mathbf{DynR}(t)$ from Eq (8). For each considered time $t_0$, each curve $\mathbf{\theta}(L, t_0)$ was smoothed using the *smooth* function of Matlab to reduce the stochastic noise before $EC_{10}(t_0)$ and $EC_{90}(t_0)$ calculation.

## Supporting information

**S1 Text. Text with additional information and calculations.**
(DOCX)

## Acknowledgments

We thank A. Colman-Lerner and L. Diambra for fruitful discussions.

## Author Contributions

**Conceptualization:** Federico Sevlever, Juan Pablo Di Bella, Alejandra C. Ventura.

**Data curation:** Federico Sevlever, Alejandra C. Ventura.

**Formal analysis:** Federico Sevlever, Alejandra C. Ventura.

**Funding acquisition:** Alejandra C. Ventura.

**Investigation:** Federico Sevlever, Juan Pablo Di Bella, Alejandra C. Ventura.

**Software:** Federico Sevlever.

**Supervision:** Alejandra C. Ventura.

**Writing – original draft:** Federico Sevlever, Alejandra C. Ventura.

**Writing – review & editing:** Federico Sevlever, Alejandra C. Ventura.

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
