## [Decision Letter · Decision Letter 0]

17 Feb 2020

Dear Dr Ventura,

Thank you very much for submitting your manuscript "Discriminating between negative cooperativity and ligand binding to independent sites using pre-equilibrium properties of binding curves" for consideration at PLOS Computational Biology. As with all papers reviewed by the journal, your manuscript was reviewed by members of the editorial board and by several independent reviewers. The reviewers appreciated the attention to an important topic. Based on the reviews, we are likely to accept this manuscript for publication, providing that you modify the manuscript according to the review recommendations.

The reviewers are quite enthusiastic about the topic and believe your results are nicely presented. They raise some concerns regarding the context and motivation for this work, which can be more strongly emphasized. In addition, the comparison to experimental data and existing studies can be expanded.

Sincerely,

Stacey Finley, Ph.D.

Associate Editor

PLOS Computational Biology

Mark Alber

Deputy Editor

PLOS Computational Biology

[LINK]

Reviewer's Responses to Questions

**Comments to the Authors:**

Reviewer #1: In the manuscript “Discriminating between negative cooperativity and ligand binding to independent sites using pre-equilibrium properties of binding curves” by Sevlever et al, the authors have investigated equilibrium and time dependent properties of dimeric receptor-ligand binding to propose an algorithmic recipe in discriminating the mechanistic details of the receptor-ligand engagement. Their proposed method was aimed at distinguishing negative cooperativity from the independent binding of a ligand to a dimeric receptor. Based on the time dependent shifting of EC50, the authors defined ‘dynamic range’ and investigated the signatory behaviors of this quantity in the independent and negative cooperative ligand binding. The signatory qualitative variation of the dynamic range with the various parameters allowed the authors to propose a new method to discriminate the two types of binding. Certainly negative cooperativity in the receptor-ligand systems is gaining a significant attention in the community due to its direct relevance in the living systems and the paper has made a significant contribution in determining the microscopic details of binding. I recommend publication of the paper after some revisions.

1. In NC model, the definition of dissociation constant (K=l/k) does not contain the 'w' term. Why? As the binding rate constant for the second binding step is 'wk', by definition the dissociation constant must have the 'w' term in it.

2. In the TC+DR algorithm, to conclude the nature of binding the authors have mentioned and used arbitrary thresholds of C. How much the results depend on the arbitrary choice of these thresholds? If the values of the thresholds were changed by a certain percentage, how much percentage change would result in the accuracy of the predictions?

3. The section about the application of their method on the previously published experimental data must be elaborated. Particularly a clear comparison of model predicted parameters and experimentally estimated parameters may be helpful. The current comparison provided in the Table S1 is not very clear.

4. Careful proofreading is necessary for typos and also for clarity in multiple places.

Reviewer #2: Sevlever et al report a nice computational work on the kinetic discrimination between negative cooperativity and binding to different types of sites, one of the most challenging topics in protein biophysics. The manuscript is generally well-written, and a concise summary of the literature is given and used to place in context the main results. The authors identify the conditions at which negative cooperativity and binding to different types of sites are indistinguishable at equilibrium. They generate a large set of kinetic parameter values fullfilling these conditions, and simulate the complete time course of the chemical species involved in the process, for different ligand concentrations. The dynamical range is then calculated and analyzed along the reaction time course. On these bases, the authors found well defined different behaviors for the time evolution of systems composed of identical sites with negative cooperativity or different types of sites without cooperativity, and propose an algorithm to distinguish between both types of binding models. Several examples from literature are used to show the efficacy of the proposed algorithm in real biological systems where experimental data is available.

In my opinion this work represents a significant contribution to knowledge in biophysics and computational biology, and I recommend its publication. However, there are some specific minor points that must be addressed in a revised version before this manuscript is suitable for publication.

1) In page 3 lines 66-69 the authors describe some biological systems displaying positive cooperativity without ligand binding events, e.g. protein folding and phospholipid melting. In these cases cooperativity can be macroscopically understood by analogy with first-order phase transitions. On the other hand, the example of DNA unwinding in lines 63-66 is somewhat controversial because enzymes are usually involved. Please discuss these points in the revised version.

2) In page 6 lines 122-123 the authors introduce the concept of “dynamical range” of the dose-response curve. This concept is closely related to the span in free ligand concentration introduced by Gregorio Weber in 1965 (Weber G, Anderson SR. Multiplicity of binding. range of validity and practical test of Adair's equation. Biochemistry. 1965; 4: 1942–1947. doi: 10.1021/bi00886a006). Please include a brief mention to this point in the revised version.

3) In Figure 6 it would be useful for the reader the addition of two new panels: (A) the dose-response curves at a given time and a selected set of parameter values, generated by stochastic simulations for 10 and 1000 receptors showing the different levels of noise; (B) the time courses of the Dynamical Range for the conditions selected in panel (A); and (C) the percentage of correct definitions using both TC and TC+DR algorithms (the one included in the current version).

4) The supplemental information contains very useful information, but it is hard to read. Please, simplify the text in the revised version.

5) Page 2 line 23, “two or more molecules” would be better than “two parties”

6) Page 3 line 58, “widely spread” would be better than “well spread”

7) Page 4 line 72, “produce no significant output” would be better than “produce no output”

8) Please revise the references. There are some problems with publication´s names, author´s names, volume and/or page numbers, etc. For the on line availability of published articles, provide the DOI instead of the URL.

Reviewer #3: The paper under consideration focusses on the problem of discriminating between negative cooperativity and ligand binding to independent sites, for which non-equilibrium (global) information is deployed. An approach along with an algorithm (with different checks) is presented towards this goal, and then used on concrete data.

The topic and analysis is interesting and will be useful to researchers who employ such models. The approach could also have broader relevance than the specific models being considered.

I think the authors can make a few changes to sharpen the paper

1. The authors need to justify a little better, why the proposed problem is interesting and of quite broad value, and consequently is well worth the study.

2. On a similar note it is also worth better situating the current problem against the backdrop of model discrimination in general in systems biology

3. I think there could be a sharper discussion of why the proposed approach is better than other approaches of this type: what does that rely on, and what implicit assumptions are being made here?

4. I think the broader value of the approach could also be a little better fleshed out: what classes of models (even within the broad class being considered) could this approach be best used for? Where are the limitations? Are there other biological motifs for which this can be fruitfully used (the authors mention adaptation, where of course, the dynamic information is central)

5. Can this approach be used systematically to infer model structures?

6. The writing and narrative is general logically laid out and clear, but in some cases could be tightened(eg heading on page 6)

7. There could be a little more discussion of the effect of varying the thresholds (page11), not just the limiting cases

8. Evaluating biological data is good. Are there examples where your analysis contradicts the existing analysis of biological data?

**Have all data underlying the figures and results presented in the manuscript been provided?**

Reviewer #1: Yes

Reviewer #2: Yes

Reviewer #3: Yes

PLOS authors have the option to publish the peer review history of their article (what does this mean?). If published, this will include your full peer review and any attached files.

Reviewer #1: No

Reviewer #2: Yes: F Luis Gonzalez Flecha

Reviewer #3: No
---

## [Decision Letter · Decision Letter 1]

6 May 2020

Dear Dr Ventura,

We are pleased to inform you that your manuscript 'Discriminating between negative cooperativity and ligand binding to independent sites using pre-equilibrium properties of binding curves' has been provisionally accepted for publication in PLOS Computational Biology.

Best regards,

Stacey Finley, Ph.D.

Associate Editor

PLOS Computational Biology

Mark Alber

Deputy Editor

PLOS Computational Biology

Reviewer's Responses to Questions

**Comments to the Authors:**

Reviewer #1: The authors have addressed all the comments/queries satisfactorily. I recommend acceptance of the manuscript for publication in the PLoS Computational Biology.

Reviewer #2: The authors have significantly improved the manuscript and addressed all the questions raised by this reviewer. Therefore, I would recommend this manuscript for publication in PLOS Computational Biology.

Reviewer #3: My comments have been addressed. My only other comment is that the authors could incorporate some part of their response to my last question in the text

**Have all data underlying the figures and results presented in the manuscript been provided?**

Reviewer #1: Yes

Reviewer #2: Yes

Reviewer #3: Yes

PLOS authors have the option to publish the peer review history of their article (what does this mean?). If published, this will include your full peer review and any attached files.

Reviewer #1: No

Reviewer #2: Yes: F Luis Gonzalez Flecha

Reviewer #3: No

---

## [Editor Report · Acceptance letter]

28 May 2020

PCOMPBIOL-D-19-02168R1 

Discriminating between negative cooperativity and ligand binding to independent sites using pre-equilibrium properties of binding curves

Dear Dr Ventura,

I am pleased to inform you that your manuscript has been formally accepted for publication in PLOS Computational Biology. Your manuscript is now with our production department and you will be notified of the publication date in due course.

With kind regards,

Laura Mallard
